# Boosting Process-Correct CoT Reasoning by Modeling Solvability of Multiple-Choice QA

## Abstract

Reasoning quality in large language models depends not only on producing correct answers but also on generating valid intermediate steps. We study this through multiple-choice question answering (MCQA), which provides a controlled setting with fixed answer options. Our analysis shows that when questions are effectively unsolvable for a model, spurious chains of thought (CoTs) are more likely to appear, leading to false positives. By estimating the solvability of each question, we uncover an intermediate regime where learning is most effective. Building on this insight, we adapt outcome-supervised reward models and reinforcement learning with group-relative advantage to incorporate solvability into their objectives. Across experiments on math and multimodal datasets, these modifications consistently yield higher rates of process-correct reasoning and, in reinforcement learning, improved answer accuracy as well. Our results highlight solvability as a key factor for reducing hallucinations and increasing reliability in CoT reasoning.

## 1 Introduction

In many applications of CoT reasoning, the generated thought process is as important as the final answer. While some tasks provide gold-standard reasoning chains that can effectively be used for supervised training (Nye et al., 2021; Dziri et al., 2023; Hochlehnert et al., 2025), most datasets lack such annotations. For these cases, correct reasoning has to be incentivized by rewards on correct final answers (Wen et al., 2025). It is known that CoTs can lead to the correct answer, despite an incorrect explanation. Grattafiori et al. (2024) note that this often occurs for questions where only a small fraction of the generated answers is correct. In this work, we investigate this observation in controlled experiments on multiple datasets. To avoid confounding factors of noisy answer extraction and matching, we focus on multiple-choice question answering. This format is popular for evaluating models and widely used training sets like NuminaMath (LI et al., 2024) contain a large fraction of multiple-choice questions. The fixed number of answer options also allows us to explicitly model the solvability of a question. We find that unsolvable questions promote false positive CoTs. Additionally, in a controlled finetuning experiment we show that there is a sweet spot of questions for which neither a small nor high fraction of CoTs lead to the correct answer. We make use of these findings by modifying the objective function of an outcome-based reward model (ORM) and by adjusting the advantage calculation of group relative reinforcement learning (RL). The proposed modifications lead to more process-correct CoTs and additionally, in the case of RL, to improved answer accuracy. Furthermore, following the argumentation of Kalai & Vempala (2024); Kalai et al. (2025) according to which LLMs are optimized to guess when uncertain, such hallucinations can be mitigated by modeling solvability in the learning objective.

## 2 Background

In chain-of-thought (CoT) reasoning, a model $\pi_\theta$ is presented with a question $q_i$ and prompted to generate an output $o_{ij}$ consisting of a thought process $t_{ij}$ and final answer $\hat{y}_{ij}$:

$$o_{ij} \sim \pi_\theta(\cdot|q_i), \quad \text{where } o_{ij} = (t_{ij}, \hat{y}_{ij}). \tag{1}$$

To ensure diverse outputs when sampling multiple CoTs per question, the token logits are divided by a positive temperature value. In this work, we use temperature 1.0 for all experiments and reported

results. Given the ground-truth answer $y_i$, the correctness of the generated answer is determined by a binary scoring function $\mathbb{1}[y_i=\hat{y}_{ij}]$ that equals 1 if $y_i = \hat{y}_{ij}$ and 0 otherwise. Because matching of open-ended answers can be ambiguous, multiple-choice question answering (MCQA) is a popular format. Each question includes a letter-indexed list of predefined answer choices $c_i$ with exactly one correct choice, and scoring reduces to exact letter matching.

**Metrics**   Given a dataset $\mathbb{D}$ with question-answer pairs $(q_i, y_i) \in \mathbb{D}$, the performance of CoT reasoning is commonly measured by answer accuracy (A-Acc). The generated answer $\hat{y}_{ij}$ is compared with the ground truth answer $y_i$ and the binary score is averaged across questions and samples:

$$\text{A-Acc} := \frac{1}{|\mathbb{D}|G} \sum_{i=1}^{|\mathbb{D}|} \sum_{j=1}^{G} \mathbb{1}[y_i=\hat{y}_{ij}]. \tag{2}$$

A CoT consists of the thought process and the final answer. The correctness of the latter can be determined by comparing it to the ground-truth answer. Because there does not exist *the one* ground-truth thought process, we employ an LLM to judge its correctness. Although this is common practice, it has to be handled with care (He et al., 2024; Hao et al., 2024; Bavaresco et al., 2025). As such, we conduct an extensive meta-evaluation of the judge by reporting correlation with human judgments, measuring performance on a synthetic dataset, and manually evaluating a subset of the judgments (Appendix D). In addition, we release all outputs verbatim to facilitate future comparisons with our work. Formally, the judge $\mathcal{J}_{LLM}$ receives the question $q_i$, a thought process $t_{ij}$, and the ground-truth answer $y_i$ as input, and returns the binary judgment:

$$\mathcal{J}_{\text{LLM}} : (q_i, t_{ij}, y_i) \to \{0, 1\}. \tag{3}$$

We then calculate the process accuracy (P-Acc) as:

$$\text{P-Acc} := \frac{1}{|\mathbb{D}_{AC}|} \sum_{q_i, t_{ij}, y_i \in \mathbb{D}_{AC}} \mathcal{J}_{\text{LLM}}(q_i, t_{ij}, y_i). \tag{4}$$

In this work, we report process accuracy solely on the subset of answer-correct CoTs, denoted as $\mathbb{D}_{AC}$. A CoT that is answer-correct but process-incorrect is referred to as *false positive*.

**Outcome-Supervised Reward Model**   An outcome-supervised reward model (ORM) is used to predict the correctness of a generated answer without access to the ground-truth (Cobbe et al., 2021). It is denoted as $\pi_\phi$ and its training objective is to minimize the binary cross entropy loss:

$$\mathcal{L}_{ij}^{\text{BCE}}(\pi_\phi) = -z_{ij} \log \pi_\phi(h_{ij}) - (1 - z_{ij}) \log(1 - \pi_\phi(h_{ij})) \tag{5}$$

where $z_{ij} = \mathbb{1}[y_i=\hat{y}_{ij}]$ is the label. The input $h_{ij}$ is the representation of the question and the sampled output $(q_i, o_{ij})$, e.g., the raw text or the LLM's last hidden state during generation. The ORM can then be used to rerank outputs at test-time or as reward estimator in reinforcement learning.

**Reinforcement Learning with Group Relative Advantage**   Recently, reinforcement learning with estimation of group relative advantage has gained renewed traction, especially in domains with verifiable rewards (Kool et al., 2019; Shao et al., 2024; DeepSeek-AI et al., 2025). The approach eliminates the complexity of training a reward model and the accompanying problems such as reward hacking or data bias. Instead, the advantage or value of an action $o_{ij}$ is determined relative to other samples for the same input. This means we sample multiple outputs per question and the reward for each output is computed as $r_{ij} = \mathbb{1}[y_i=\hat{y}_{ij}]$, where $\hat{y}_{ij}$ is *null* if no answer can be extracted from $o_{ij}$, e.g. due to incorrect answer format. The advantage is then calculated relative to the rewards of the other samples. Specifically, the formulations of the GRPO (Shao et al., 2024) and DrGRPO (Liu et al., 2025) variants are:

$$A_{ij}^{\text{GRPO}} = \frac{1}{\sigma(\boldsymbol{r}_i)} \left( r_{ij} - \frac{1}{G} \sum_{k=1}^{G} r_{ik} \right) \quad \text{and} \quad A_{ij}^{\text{DrGRPO}} = r_{ij} - \frac{1}{G} \sum_{k=1}^{G} r_{ik}, \tag{6}$$

respectively, where $\boldsymbol{r}_i$ is the reward vector and $\sigma(\cdot)$ returns the standard deviation. The policy gradient for a single question, simplified here without the standard PPO clipping term, becomes:

$$\nabla_\theta J(\theta) \approx \frac{1}{G} \sum_{j=1}^{G} A_{ij} \nabla_\theta \log \pi_\theta(o_{ij} \mid q_i), \tag{7}$$

where the advantage $A_{ij}$ of a sample $o_{ij}$ is computed using GRPO or DrGRPO as defined above.

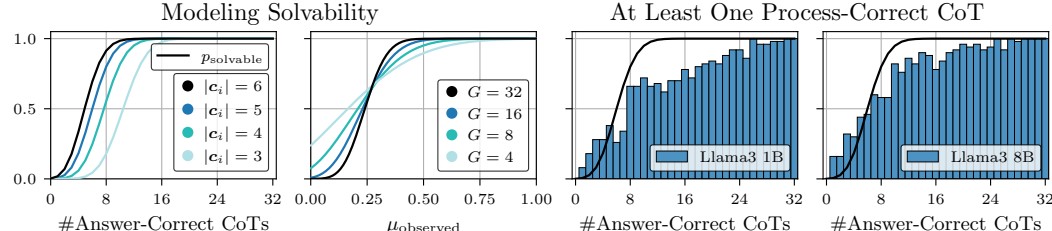

Figure 1: **Modeling Solvability:** The probability that a question is solvable by a given model, as defined by Equation 11. (Left) Varying number of answer options for the multiple-choice question. (Right) Varying number of sampled CoTs per question. **At Least One Process-Correct CoT:** Fraction of questions where at least one of the 32 generated CoTs is process-correct. Questions are from the AQuA dataset (five choices) and CoTs are sampled with Llama3 1B (left) and Llama3 8B (right).

## 3 SOLVABILITY OF MULTIPLE-CHOICE QUESTIONS

We model the binary outcome of whether a CoT $o_{ij}$, sampled from model $\pi_\theta$, correctly answers a question $q_i$ as a Bernoulli random variable:

$$\mathbb{E}_{o_{ij}\sim\pi_\theta(\cdot|q_i)}\left[\mathbb{1}\left[y_i=\hat{y}_{ij}\right]\right]=\mu_{\text{true}}^\theta(q_i). \tag{8}$$

The true parameter $\mu_{\text{true}}^\theta(q_i)$ is unobservable, but can be estimated by sampling $G$ outputs:

$$\mu_{\text{observed}}^\theta(q_i)=\frac{1}{G}\sum_{j=1}^{G}\mathbb{1}[y_i=\hat{y}_{ij}]. \tag{9}$$

Using a uniform prior Beta$(1, 1)$ and the observed success rate $\mu_{\text{observed}}^\theta(q_i)$ as likelihood, the posterior distribution for $\mu_{\text{true}}^\theta(q_i)$ is given by Beta$(\alpha_i, \beta_i)$ with the parameters:

$$\alpha_i=1+G\mu_{\text{observed}}^\theta(q_i)\quad\text{and}\quad\beta_i=1+G\left(1-\mu_{\text{observed}}^\theta(q_i)\right), \tag{10}$$

which represent the success and failure counts, respectively. We define a question as solvable by the model if the model's true performance exceeds random guessing: $\mu_{\text{true}}^\theta(q_i)>\mu_{\text{random}}(q_i)$. The random guessing baseline for a multiple-choice question is given by: $\mu_{\text{random}}(q_i)=\frac{1}{|c_i|}$. Using this information, we compute the probability that a question is solvable for the model as the survival function of the Beta distribution:

$$p_{\text{solvable}}^\theta(q_i)=p(\mu_{\text{true}}^\theta(q_i)>\mu_{\text{random}}(q_i))=\int_{\mu_{\text{random}}(q_i)}^1\text{Beta}(\mu;\alpha_i,\beta_i)d\mu. \tag{11}$$

The left section of Figure 1 illustrates $p_{\text{solvable}}^\theta(q_i)$ for varying number of answer choices $|c_i|$ and varying number of samples $G$. When only a small fraction of CoTs yield the correct answer, the model's probability of solving the question approaches zero. As the number of answer-correct CoTs increases, solvability rises exponentially before converging to unity. Both the onset and the inflection point depend on the number of answer choices. The more answer choices a question offers, the smaller the proportion of correct CoTs that is required to achieve solvability. The steepness of the increase depends on the number of sampled CoTs — more samples provide a clearer distinction between solvable and unsolvable questions.

### 3.1 SOLVABILITY AND PROCESS-CORRECTNESS

Intuitively, if a question is not solvable for a model, the model should not be able to generate a CoT with correct thought process. We empirically verify this intuition in Figure 1 (right section). There, the questions in the math reasoning dataset AQuA (Ling et al., 2017) are categorized by the number of answer-correct CoTs, generated by Llama3 1B and Llama3 8B (Grattafiori et al., 2024). The $p_{\text{solvable}}^\theta(q_i)$ line closely follows the empirical data in the bar chart, showing it is a good predictor of whether the model is able to generate a correct thought process for a given question. By incorporating this probability into the training of an outcome reward model (Section 4) and advantage calculation of reinforcement learning (Section 5), we expect to boost the ability to identify and generate process-correct CoTs, respectively.

Table 1: **Process-Accuracy (P-Acc)** using different methods of CoT scoring. The task is to score multiple candidate CoTs that all lead to the correct answer. The highest scoring CoT is then evaluated for process-correctness. This is done for 200 questions of the three multiple-choice QA datasets. Oracle gives the upper bound because not every candidate set contains a process-correct CoT. Outcome-supervised reward models (ORM) are trained with three different random seeds and mean±std is reported. Nominal best values are bold.

| | Llama3 1B | | | Llama3 8B | | |
|---|---|---|---|---|---|---|
| | **AQuA** | **MATH** | **GSM8K** | **AQuA** | **MATH** | **GSM8K** |
| Oracle | 79.5 | 80.0 | 93.5 | 96.0 | 92.0 | 98.0 |
| Random | 47.0 | 45.7 | 66.0 | 81.5 | 63.3 | 90.2 |
| Shortest | 47.0 | 51.5 | 65.0 | 87.0 | 75.0 | 94.0 |
| Longest | 26.0 | 27.0 | 47.5 | 53.0 | 36.0 | 61.5 |
| CoPS (Wang et al., 2025) | 52.0 | 54.5 | 69.5 | 61.5 | 49.5 | 72.5 |
| Faithfulness (Paul et al., 2024) | 37.5 | 39.5 | 58.0 | 71.0 | 56.0 | 87.0 |
| Answer Confidence | 57.5 | 48.5 | 76.5 | 84.0 | 73.5 | 92.0 |
| ORM | 67.3 ±0.8 | 64.5 ±1.5 | 87.8 ±1.0 | 90.3 ±0.2 | 81.0 ±0.8 | 95.7 ±0.2 |
| MCQ-ORM (ours) | **70.0** ±0.7 | **65.7** ±0.2 | **88.7** ±0.5 | **92.0** ±0.4 | **83.5** ±0.4 | **96.2** ±0.5 |

## 4 BOOSTING PROCESS-CORRECTNESS AT TEST-TIME

A common technique to improve answer accuracy at test-time is to sample multiple CoTs for a question and select the final answer by majority vote (Wang et al., 2023b). This means that there are multiple candidate CoTs that lead to the majority-voted answer. In this section, we discuss the task of selecting the candidate CoT that is most likely process-correct. Uesato et al. (2023) use the score of an ORM to select the most promising CoT among the candidates. They show that this improves average process-correctness in comparison to random selection. Their ORM is trained with binary outcome labels where answer-correct CoTs have label 1, and 0 otherwise. Instead, we incorporate the probability that a question is solvable (Equation 11) into the ORM objective (Equation 5):

$$z_{ij} = \begin{cases} p_{\text{solvable}}^{\theta}(q_i), & \text{if } \hat{y}_{ij} = y_i \\ 0, & \text{otherwise.} \end{cases} \tag{12}$$

This gives lower weight to CoTs that are likely *false positive* and therefore should receive a lower score when ranking the candidate CoTs during test-time. We call this modification MCQ-ORM to reflect the incorporation of solvability of multiple-choice questions.

### 4.1 EXPERIMENTS

We train the proposed MCQ-ORM and estimate its accuracy in selecting a process-correct CoT among multiple answer-correct CoTs. We compare it to the unmodified ORM and other baselines. As base models we use Llama3 with 1B and 8B parameters. The training and development set for the reward models are sourced from the 97.5k training questions of AQuA. We report process accuracy for all methods on three math reasoning datasets for both base models.

**Reward Model Training** Given a base model, we sample 32 CoTs for each of the 97.5k training questions, resulting in 3M training and 32k development instances for the reward model. A training instance consists of the base model's last hidden state as input and the appropriate outcome-based label (see Equation 5 for the unmodified ORM and Equation 12 for our proposed MCQ-ORM). The architecture is a feed-forward neural network with two hidden layers and sigmoid activations. Additional hyperparameters are optimized individually for each reward model (Appendix C). Both ORM variants are trained with three different random seeds, and early stopping is based on cross-entropy loss of the development set.

**Evaluation** In addition to AQuA (five choices), we also report results on the MATH (Hendrycks et al., 2021) and GSM8K (Cobbe et al., 2021) datasets. Both are modified to follow the MCQA format with four choices and six choices, respectively (Zhang et al., 2024). For each test question,

we sample 32 CoTs and determine the predicted answer by majority vote. The subset of CoTs that lead to the majority answer is the candidate set. The task is to select one CoT among the candidates that is most likely process-correct. Each method is evaluated on the same 200 questions from each dataset ranging from 6 to 32 candidate CoTs. Because it is not guaranteed that at least one candidate is process-correct, we also report oracle results as an upper bound.

## 4.2 RESULTS

The results in Table 1 show that both base models are generally capable of generating process-correct CoTs. The large gap between the random baseline and the oracle shows substantial room to test the capabilities of the considered selection methods. The CoPS (Wang et al., 2025) and faithfulness (Paul et al., 2024) baselines are making use of early answer probing. After each reasoning step, the model is forced to decode the correct answer letter and its token probability is recorded in a vector. This early answer probability vector is then used to draw conclusions about the reasoning process. CoPS estimates the quality of a CoT by considering the average probability of early answers and their increase over time. The CoT faithfulness metric was developed to measure the alignment of the model's internal with its external textual reasoning. It is defined as the area over the curve of the early answering probability vector. We can see that both of these metrics are outperformed by a simple baseline that ranks CoTs by final answer confidence, i.e., the probability assigned to the correct answer letter. Using the score of a reward model to select the best CoT largely outperforms the aforementioned baselines. Our proposed MCQ-ORM that takes the solvability of a question into account consistently outperforms the unmodified ORM across considered datasets and base models. Although the effect size is small, a random permutation test (Appendix B) shows that the results are overall significant.

## 5 REINFORCEMENT LEARNING WITH ADJUSTED ADVANTAGE

We start with an analysis of advantage values calculated by GRPO and DrGRPO. The plots on the left and middle of Figure 2 show that a sample $o_{ij}$ with positive reward $r_{ij} = 1$ gets the highest individual advantage if all other samples in the group received a negative reward. Comparing this to the right section of Figure 1, we see that these samples correspond to CoTs that are most likely process-incorrect. To further investigate the impact of this advantage shape, we conduct a controlled experiment where we estimate the learning potential of question-CoT pairs, depending on the number of answer-correct CoTs in the group.

## 5.1 ESTIMATING LEARNING POTENTIAL

We sample 32 CoTs for each question in the respective training set. The questions are then categorized into buckets based on the number of answer-correct CoTs in the group. We then randomly select a subset of the questions in a bucket. For each of the questions in the subset, we randomly select exactly one of its answer-correct CoTs. Formally, a finetuning dataset for bucket $b$ is:

$$\mathbb{D}_b^{FT} = \{(q_i, o_{ij}) \mid q_i \sim Q_b, j \sim \{k : r_{ik} = 1\}\}, \text{ where } Q_b = \{q_i \mid \sum_{k=1}^{G} r_{ik} = b\}. \quad (13)$$

For each bucket, such a dataset with 2k instances is sampled and used to finetune the base model. We then evaluate the finetuned models on the development set of the respective dataset and measure the improvement in answer-accuracy over the base model. Figure 3 shows the results for AQuA (Ling et al., 2017), MedMCQA (Pal et al., 2022) and SocialIQA (Sap et al., 2019) using Llama2 7B (Touvron et al., 2023). Each experiment is repeated five times with different random seeds and the mean is depicted. The seed affects the subset selection and the order of training batches. Buckets with less than 2k questions are merged with their neighbor bucket. For all datasets, the observed distribution of accuracy improvement is left-skewed, with a linear decrease to the right and a steep increase on the left. The position of the distribution mode varies depending on the dataset. In the following, we derive a simple model that describes the observed accuracy improvement of question-CoT pairs based on bucket membership and number of answer choices.

Given a question and sampled CoTs, we seek to model the learning potential (LP) of a pair $(q_i, o_{ij})$.

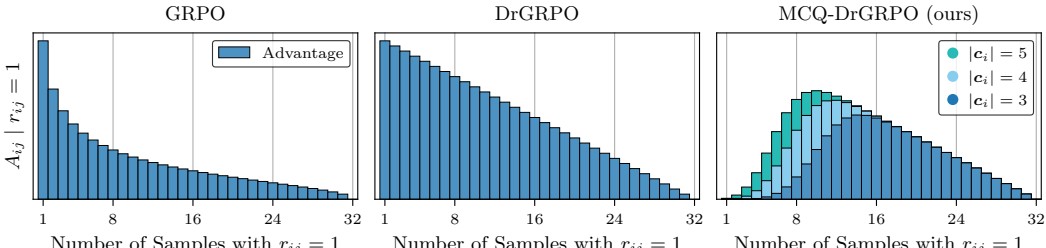

Figure 2: Advantage values of a single CoT with positive reward. 32 CoTs are sampled for each question and the x-axis denotes the number of answer-correct (positive reward) CoTs in a group. MCQ-DrGRPO down-weights CoTs that are generated for unsolvable questions. The probability that a multiple-choice question is unsolvable for the model depends on the number of choices $|c_i|$. The values on the y-axis are omitted to allow visual comparison across methods. During training the relative differences between groups are important.

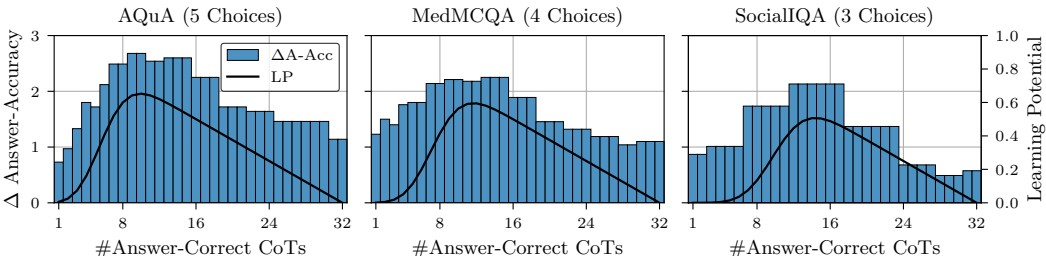

Figure 3: We sample 32 CoTs for each question in the respective training set. Questions are then categorized into buckets based on the number of answer-correct CoTs. We randomly sample questions from each bucket and pair them with exactly one of their answer-correct CoTs. We finetune the base model on these 2k instances and report the increase in answer accuracy over the base model on a held out development set. Experiments are repeated five times with different random seeds. Learning potential (LP) predicts relative increase in answer accuracy based on bucket membership.

Questions that are trivially solved by the model offer minimal informational gain, as they lack novelty with respect to the model's existing knowledge. In contrast, questions that the model fails to answer correctly contain maximal novel information. We formalize the probability that a question provides novel information as the fraction of incorrect answers:

$$p_{\text{novel}}^{\theta}(q_i) = \frac{1}{G} \sum_{j=1}^{G} \mathbb{1}[y_i \neq \hat{y}_{ij}]. \tag{14}$$

However, as seen in Figure 3, a counteracting mechanism limits the model to learn from overly novel inputs. Specifically, when a question exceeds the model's current capabilities, the learning signal becomes noisy or cannot be effectively utilized. We capture this trade-off through the following formulation:

$$\text{LP}(q_i, o_{ij}) = p_{\text{novel}}^{\theta}(q_i)\, p_{\text{solvable}}^{\theta}(q_i). \tag{15}$$

The line in Figure 3 shows that this estimation of learning potential aligns well with the observed improvement in accuracy. We will use this finding to adjust the advantage calculation in order to prefer instances with high learning potential.

## 5.2 ADVANTAGE CALCULATION ADJUSTED BY SOLVABILITY

Using equality $\mathbb{1}[y_i = \hat{y}_{ij}] = 1 - \mathbb{1}[y_i \neq \hat{y}_{ij}]$, we can rearrange the DrGRPO advantage calculation (Equation 6) of a sample with positive reward to be equal to the novelty formulation in Equation 14. Applying our findings that there is a trade-off between novelty and solvability, we propose solvability-adjusted DrGRPO for multiple-choice questions:

$$A_{ij}^{\text{MCQ-DrGRPO}} = p_{\text{solvable}}^{\theta}(q_i) A_{ij}^{\text{DrGRPO}} \tag{16}$$

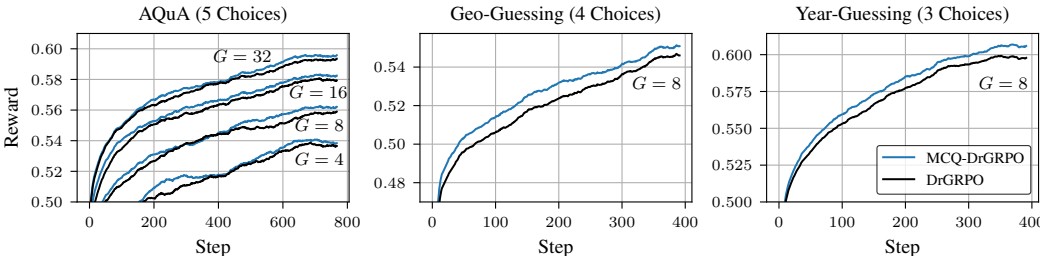

Figure 4: The average reward during RL training with DrGRPO and MCQ-DrGRPO. The math reasoning dataset AQuA is used to train Llama3 1B and geo/year-guessing datasets are used to train multimodal Aya 8B. First graph also shows ablations for different numbers of sampled CoTs per question. Each model is trained with three different random seeds.

The plot on the right in Figure 2 shows that this calculation focuses advantage mass on samples with high learning potential. CoTs sampled for unsolvable questions, and hence likely process-incorrect, are downweighted compared to GRPO and DrGRPO.

## 5.3 Reinforcement Learning Experiments

We perform reinforcement learning experiments using proximal policy optimization (Schulman et al., 2017) with group relative advantage estimation (Shao et al., 2024). Specifically, we compare the advantage estimation of DrGRPO (Equation 6) to our proposed MCQ-DrGRPO (Equation 16) that incorporates the solvability of a multiple-choice question into the advantage calculation. We use KL penalty and remove output length bias Liu et al. (2025). See Appendix C for more details of the implementation. Besides the answer accuracy (A-Acc) we also report the process accuracy (P-Acc) which is the average correctness of the thought processes that lead to a correct answer (Section 2).

**Math Reasoning** We use AQuA (five choices) as the training set and report evaluation metrics on two additional datasets. These are MATH and GSM8K, both modified to follow the MCQA format with four choices and six choices, respectively (Zhang et al., 2024). As base models, we use Qwen2.5 1.5B (Qwen et al., 2025) and Llama3 1B because it is not saturated on the considered datasets and the relatively small size allows us to conduct additional ablation experiments. We sample 32 CoTs per question during training.

**Multimodal Reasoning** Due to the lack of large-scale multimodal reasoning datasets that are *not* math-related, we construct two novel MCQA datasets. One asks for the geographic region (four choices) in which an image was taken and the other for the year (three choices) when it was taken. These tasks require the model to analyze different aspects of the image and combine it with general knowledge to draw a conclusion. Both datasets have 93k training instances, and a development and test set of size 3.5k each. See Appendix A for more details and download link. We use mulimodal Aya 8B (Dash et al., 2025) as the base model and sample 8 CoTs per question during training.

## 5.4 Results

The plots in Figure 4 show average rewards during training. Our proposed MCQ-DrGRPO achieves consistently higher rewards than the DrGRPO baseline. The experiments are repeated three times with different random seeds, affecting data ordering and token sampling. Table 2 and Table 3 show the process and answer accuracy, evaluated on three datasets. Both methods improve not only answer accuracy over the base model, but also process accuracy. This confirms recent findings that reinforcement learning with verifiable rewards (RLVR) implicitly optimizes the correctness of the thought process (Wen et al., 2025). Our proposed MCQ-DrGRPO consistently outperforms Dr-GRPO with larger effect sizes for process accuracy. This shows the effectiveness of downsizing advantage values of unsolvable questions. A randomized permutation test across seeds and datasets shows that MCQ-DrGRPO achieves significantly higher process and answer accuracy than the baseline. Table 4 shows that the results also hold for multimodal reasoning and for an out-of-domain setting, where the model is trained on geo-guessing and evaluated on year-guessing.

Table 2: **Process-Accuracy (P-Acc)** and **Answer-Accuracy (A-Acc)** for CoTs sampled by Llama3 1B and RL-tuned derivations. The AQuA dataset is used for RL and we additionally evaluate on the MATH and GSM8k datasets, both adapted to the multiple-choice format. The RL training is repeated three times with different random seeds and mean±std is reported. P-Acc and A-Acc are calculated using 200 and 2k questions, respectively. Nominal best values are bold.

| | AQuA | | MATH | | GSM8K | |
|---|---|---|---|---|---|---|
| | **P-Acc** | **A-Acc** | **P-Acc** | **A-Acc** | **P-Acc** | **A-Acc** |
| Base Model (Llama3 1B) | 47.0 | 41.9 | 45.7 | 45.1 | 66.0 | 55.6 |
| DrGRPO | 63.7 ±1.6 | 58.9 ±0.0 | 46.3 ±1.5 | 62.5 ±0.1 | 71.2 ±0.8 | 77.4 ±0.1 |
| MCQ-DrGRPO (ours) | **65.0** ±0.7 | **59.5** ±0.3 | **50.0** ±1.8 | **62.6** ±0.1 | **73.5** ±1.4 | **78.2** ±0.3 |

Table 3: **Process-Accuracy (P-Acc)** and **Answer-Accuracy (A-Acc)** for CoTs sampled by Qwen2.5 1.5B and RL-tuned derivations. The AQuA dataset is used for RL and we additionally evaluate on the MATH and GSM8k datasets, both adapted to the multiple-choice format. The RL training is repeated three times with different random seeds and mean±std is reported. P-Acc and A-Acc are calculated using 200 and 2k questions, respectively. Nominal best values are bold.

| | AQuA | | MATH | | GSM8K | |
|---|---|---|---|---|---|---|
| | **P-Acc** | **A-Acc** | **P-Acc** | **A-Acc** | **P-Acc** | **A-Acc** |
| Base Model (Qwen2.5 1.5B) | 70.0 | 52.0 | 68.5 | 64.7 | 80.0 | 66.3 |
| DrGRPO | 81.3 ±2.1 | 70.6 ±0.1 | 76.0 ±2.2 | 71.0 ±0.2 | 86.2 ±1.8 | 77.0 ±0.2 |
| MCQ-DrGRPO (ours) | **82.0** ±1.5 | **71.1** ±0.2 | **76.3** ±1.3 | **72.1** ±0.3 | **87.8** ±3.2 | **81.4** ±0.4 |

## 5.5 ANALYSIS

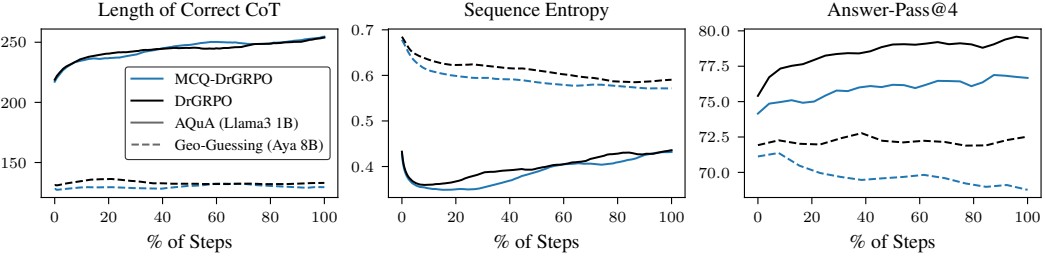

Figure 5: Additional metrics recorded during the reinforcement learning experiments. **Length of Correct CoT:** Average number of tokens in an answer-correct CoT. **Sequence Entropy:** Summed token entropy for a CoT sequence, normalized by length. **Answer-Pass@4:** Percentage of questions with at least one answer-correct CoT among four samples.

In Figure 6, we vary the number of CoTs sampled per question during training. Each data point represents the average of nine values: three models trained with different random seeds, each evaluated on the three math reasoning datasets. The improvement in answer accuracy is consistent across number of samples and the gap in process accuracy is widening with more samples. This can be explained by the clearer identification of unsolvable questions with an increasing number of samples $G$, as shown in Figure 1 (second graph). To better understand MCQ-DrGRPO's impact, we track additional metrics during training. The plot on the left of Figure 5 shows that the average length of answer-correct CoTs is comparable for both models, ruling out length bias as an expla-

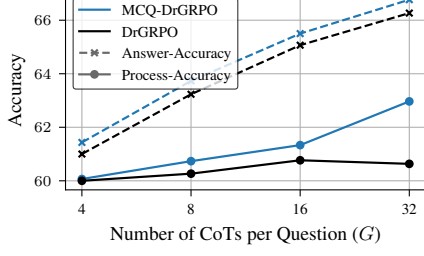

Figure 6: Results for varying number of CoTs per question. Average across three datasets and three random seeds. Relates to Table 2 and Figure 4 (left).

Table 4: **Process-Accuracy (P-Acc)** and **Answer-Accuracy (A-Acc)** for CoTs sampled by the base model (Aya 8B) and RL-tuned derivations. Training on geo-guessing and testing on year-guessing (and vis versa) constitutes out-of-domain evaluation. The RL training is repeated three times with different random seeds and mean±std is reported. P-Acc and A-Acc are calculated using 200 and 3.5k questions, respectively. Nominal best values are bold.

| | RL Dataset | Geo-Guessing | | Year-Guessing | |
|---|---|---|---|---|---|
| | | P-Acc | A-Acc | P-Acc | A-Acc |
| Base Model | | 47.0 | 46.4 | 48.5 | 41.9 |
| DrGRPO | Geo-Guessing | 50.2 ±2.5 | 55.6 ±0.1 | 56.8 ±3.9 | **42.6** ±0.2 |
| MCQ-DrGRPO (ours) | | **55.2** ±0.6 | **56.0** ±0.3 | **58.3** ±3.1 | **42.6** ±0.1 |
| DrGRPO | Year-Guessing | 57.7 ±4.2 | 46.0 ±0.2 | 52.7 ±1.0 | 52.8 ±0.3 |
| MCQ-DrGRPO (ours) | | **59.0** ±0.7 | **46.9** ±0.3 | **57.5** ±2.9 | **53.4** ±0.4 |

nation for the difference in process accuracy. The plot in the middle reveals that the sequence entropy is lower for CoTs generated by the MCQ-DrGRPO model. This means that the output distribution learned by MCQ-DrGRPO is sharper than that learned by DrGRPO. This observation is supported by the graph on the right that shows the percentage of questions that are answered correctly by at least one out of four sampled CoTs. DrGRPO outperforms MCQ-DrGRPO in this metric, which means that the variance of answers is higher using DrGRPO. This aligns with recent work which finds that RL with verifiable rewards does not truly learn new things, but sharpens the distribution toward answer-correct CoTs (Yue et al., 2025). MCQ-DrGRPO effectively prioritizes reliable training signal over diverse but potentially noisy signal. This trade-off results in a sharpened distribution that generates correct CoTs more consistently.

# 6 RELATED WORK

**CoT Process-Correctness**   The evaluation of CoT reasoning is primarily focused on answer correctness (Wei et al., 2022; Wang et al., 2023b; Fu et al., 2023; Liu et al., 2023; DeepSeek-AI et al., 2025). Because a correct answer does not imply correct reasoning (Wang et al., 2023a), evaluating the process is of interest for many applications (Singhal et al., 2022; Blair-Stanek et al., 2023; Macina et al., 2023). Process correctness in compositional reasoning tasks can often be verified by a parser (Cobbe et al., 2021; Willig et al., 2022; Lyu et al., 2023; Xu et al., 2024), but most natural language tasks require human annotators (Collins et al., 2022; Zelikman et al., 2022; Uesato et al., 2023; Mondorf & Plank, 2024). Only recently have studies explored training models (Golovneva et al., 2023; Prasad et al., 2023) or using LLMs (He et al., 2024; Hao et al., 2024; Bavaresco et al., 2025) to judge the correctness of reasoning chains. Uesato et al. (2023) show that process reward models (PRM), learned from human annotations, are improving process correctness at test time. Recently, PRM training moved away from human annotations (Lightman et al., 2024) towards implicit step-level feedback derived from final answer correctness (Yuan et al., 2025; Wang et al., 2024).

**Advantage Calculation and Data Difficulty**   There are many works that modify the advantage calculation of GRPO (Shao et al., 2024). DrGRPO (Liu et al., 2025) drops the normalization by standard deviation in order to reduce the "question-level difficulty bias". Other works incorporate an entropy reward to encourage more diverse CoTs Zhang et al. (2025b); Cheng et al. (2025) or penalize uncertainty (Chen et al., 2025). Zhang & Zuo (2025) reweigh the advantage based on question difficulty, calculated as the fraction of correct answers. They increase the weight of CoTs that correctly answer a difficult question. This contrasts the trade-off between difficulty and novelty (Swayamdipta et al., 2020).

# 7 CONCLUSION

We explicitly modeled the ability of an LLM with CoT reasoning to solve a certain multiple-choice question. To this end, a group of sampled CoTs is used to estimate the probability that the true

performance of the LLM exceeds random guessing. We incorporated the estimated solvability of a question into the objective of an outcome-based reward model and reinforcement learning with group-relative advantage estimation. Experiments on different base models and datasets showed improved process accuracy of emitted CoTs, and additionally improved answer accuracy in the case of RL. Supporting experiments confirmed that answer-correct CoTs from groups with few correct answers are more likely to be process-incorrect and provide noisy learning signal. The method does not introduce additional hyperparameter and requires negligible computation overhead. Taking solvability into account should become the default when using multiple-choice questions for reasoning training. Although it is not straightforward to adapt our solvability formulation to open-answer tasks, one possible direction is reformulation to MCQA (Zhang et al., 2025a). Our work highlights that unsolvable questions are a source of *false positive* CoTs and a method to mitigate this phenomenon in the MCQA setting.

## REPRODUCIBILITY

The supplementary material includes training code, data and outputs of our experiments. We will additionally release model checkpoints in a github repository. The Appendix lists model hyperparameters and used prompts.

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

APPENDIX

# A  GEO-GUESSING AND YEAR-GUESSING MCQA

We introduce two multiple-choice question answering (MCQA) datasets for multimodal reasoning. Both are derived from the YFCC100M dataset (Thomee et al., 2016) that provides 100M images from Flickr, partially annotated with metadata like tags, title and geolocation. In the following, we describe the construction of the Geo-Guessing MCQA and Year-Guessing MCQA datasets. For Geo-Guessing MCQA, we start with MP16 (Larson et al., 2017), a YFCC100M subset of 7M images that are tagged with geographic coordinates in the form of latitude and longitude. We use Nominatim (OpenStreetMap Foundation, 2009) to translate the coordinates into a textual description of the region where an image was taken. We downloaded each image in the highest available resolution, sorted them by resolution and selected the top 100k instances. These instances are then split into 92k training, 4k development and 4k test instances. The process for Year-Guessing MCQA is the same as for geo, except that we start with DEW (Müller et al., 2017), also a subset of YFCC100M, and extract the year an image was taken from DEW's additional annotation. They derived the year an image was taken from user-provided metadata like title, description and tags. An example instance of Geo-Guessing MCQA is shown in Figure 7 and of Year-Guessing MCQA in Figure 8.

## A.1  GENERATING ANSWER CHOICES

Generating incorrect answer choices (distractors) for MCQA should be handled with care to avoid exposing the correct answer via subtle bias. For example, generating distractor years via symmetric error, e.g., $uniform(year - distance, year + distance)$, let's a model learn to predict the median value as correct. On the other hand, to have strong distractors, we need to generate choices that are close to the correct answer. To this end, we designed an algorithm that is not biased by the correct value and offers parameters to regulate the maximum distance to the correct value as well as the minimum distance between answers. The latter is needed to increase distinguishability. The algorithm is laid out in Algorithm 1 and the parameters for Year-Guessing are set to $n = 2, d = 20, s = 4$. The algorithm first shifts the center in a random direction by a maximum distance of $d//2$. Then it samples the distractors uniformly around the shifted center, with a maximum distance of $d//2$. The process is repeated until $n$ distractors are generated that all satisfy the minimum intra-distance constraint. For Geo-Guessing we modify the algorithm slightly. The correct value is a pair of latitude and longitude, the value is shifted by a distance in kilometers and random bearing ($uniform(1, 360)$). Distractors that have the same textual representation (by Nominatim) as any of the existing answers are considered equal, even if they have different coordinates. The parameters for Geo-Guessing are $n = 3, d = 2000, s = 5$.

---

**Algorithm 1** Generate Distractor Values with Distance Constraints

---

**Require:** correct value $a$, number of distractors $n$, maximum distance to correct value $d$, minimum distance between answers $s$

1: $shifted \leftarrow \text{RANDOMINT}(a - d//2, a + d//2)$
2: $answers \leftarrow \{a\}$
3: **while** $|answers| < n + 1$ **do**
4:     $distractor \leftarrow \text{RANDOMINT}(shifted - d//2, shifted + d//2)$
5:     $valid \leftarrow$ **true**
6:     **for each** $value \in answers$ **do**
7:         **if** $|value - distractor| < s$ **then**
8:             $valid \leftarrow$ **false**
9:         **end if**
10:    **end for**
11:    **if** $valid$ **then**
12:       $answers \leftarrow answers \cup \{distractor\}$
13:    **end if**
14: **end while**
15: **output** $answers$

---

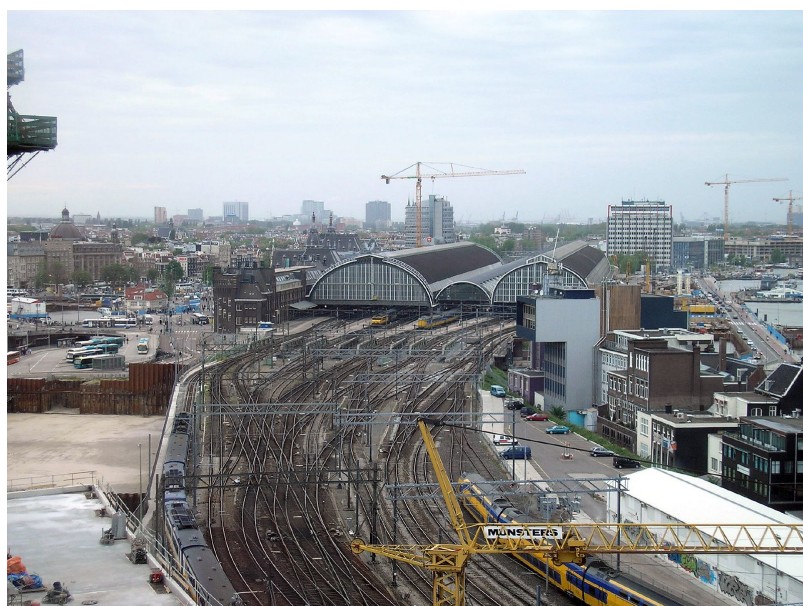

**Question:** Can you deduce the country, region and city shown in this photograph?
**Options:** A. Amsterdam, North Holland, Netherlands B. Scotland, United Kingdom
C. High Peak, Derbyshire, East Midlands, England, United Kingdom D. Risør, Agder, Norway
**Gold Answer:** A

**DrGRPO Chain-of-Thought:**
This image shows a bustling train station with multiple tracks, cranes, and urban buildings around it. The architecture and style of the buildings, along with the presence of trains and trams, suggest a European city with a well-developed public transport system. The architecture strongly hints at a city in the Netherlands, particularly Amsterdam, known for its canal system and historic buildings. The presence of cranes and the layout of the train station are consistent with the infrastructure found in major Dutch cities. Additionally, the signage in the image includes the name "Munsters," which is a local landmark in Amsterdam. The surrounding urban landscape and the style of the buildings further support this deduction.
**Judge Label:** Incorrect

**MCQ-DrGRPO Chain-of-Thought:**
This image shows a busy railway station with multiple tracks, a large archway, and urban development in the background. The presence of trains, cranes, and a variety of buildings suggests a major transportation hub in a densely populated area. The architecture and infrastructure styles are consistent with those found in the Netherlands, particularly in Amsterdam. The stations layout and the surrounding buildings resemble those found in Amsterdam's central train station and its surrounding areas. Together, these elements strongly indicate that the photograph was taken in Amsterdam, in the province of North Holland, within the Netherlands.
**Judge Label:** Correct

Figure 7: An instance of the Geo-Guessing MCQA dataset with two answer-correct CoTs generated by DrGRPO and MCQ-DrGRPO, respectively. The CoTs are evaluated for process-correctness by an LLM judge. The text marked in red is a hallucination flagged by the judge.

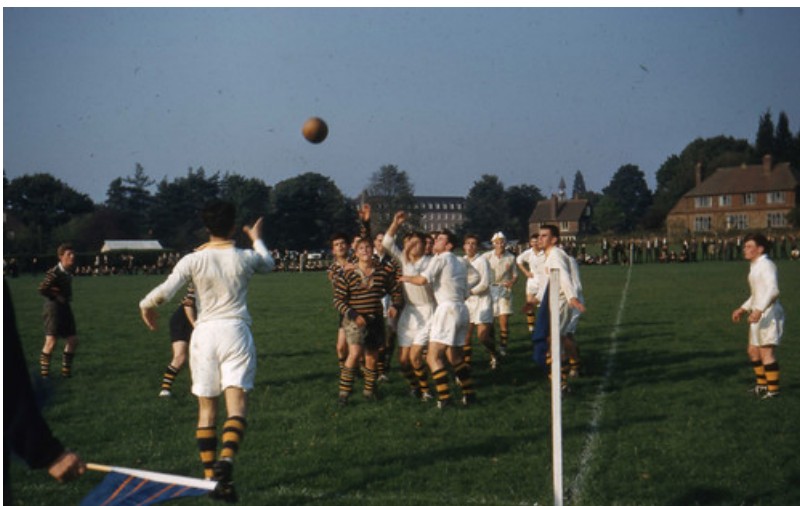

**Question:** From what you can observe, during which period was this image likely captured?
**Options:** A. 1953 B. 1961 C. 1945
**Gold Answer:** B

**DrGRPO Chain-of-Thought:**
This image appears to depict a rugby match, given the attire of the players and the equipment used. The style of the uniforms and the overall aesthetic of the scene suggest it might be from a post-World War II era, as the sport saw a resurgence in popularity during the 1940s and 1950s. The image seems to capture the spirit of the game during this time, with players engaged in the sport and a crowd gathered to watch. The presence of spectators in uniforms and the layout of the field are indicative of organized, amateur, or possibly collegiate-level matches that were common during this period.
**Judge Label:** Incorrect

**MCQ-DrGRPO Chain-of-Thought:**
The image captures a rugby match with players wearing traditional uniforms and a clear emphasis on teamwork and physicality. Considering the fashion and the context of the sport, it seems that this image was likely captured in the mid-20th century. The attire of the players, particularly the distinctive black and white striped jerseys and the long trousers, aligns with the style from the 1950s and 1960s. Additionally, the quality and composition of the image suggest it was taken during a time when photography technology was advancing but not yet as sophisticated as today's high-definition cameras. Based on these observations, the most likely period for this image is around 1961.
**Judge Label:** Correct

Figure 8: An instance of the Year-Guessing MCQA dataset with two answer-correct CoTs generated by DrGRPO and MCQ-DrGRPO, respectively. Both models are trained with the Geo-Guessing dataset. The CoTs are evaluated for process-correctness by an LLM judge. The text marked in red is a hallucination flagged by the judge.

## B STATISTICAL SIGNIFICANCE

We perform a stratified random permutation test to establish the significance of the reported results. For each group of experiments, we pool the accuracy values of both methods across all seeds and randomly reassign them to two groups. We repeat this permutation process 100k times within each dataset independently, then compute the mean difference across datasets for each permutation. The two-tailed p-value is the proportion of permutations where the absolute value of the permuted mean difference is greater than or equal to the absolute value of the observed mean difference. The following p-values indicate the statistical significance of the improvement when using the proposed MCQ variants compared to the baseline ORM and RL methods. Table 1 Llama3-1B P-Acc p-value: 0.0115; Table 1 Llama3-8B P-Acc p-value: 0.0010; Table 2 Llama3-1B 32 Samples P-Acc p-value: 0.0077, A-Acc p-value: 0.0008; Table 4 Aya-8B P-Acc p-value: 0.0249, A-Acc p-value: 0.0023.

Table 5: Hyperparameter for reward model training. Hyperparameter were selected by cross-entropy loss on the development set.

| | Llama3 1B | | Llama3 8B | |
|---|---|---|---|---|
| | **ORM** | **MCQ-ORM** | **ORM** | **MCQ-ORM** |
| Batch Size | | 512 | | |
| Dropout | | 0.0 | | |
| Gradient Norm | | 1.0 | | |
| Learning Rate | | 0.0001 | | |
| Weight Decay | | 0.001 | | |
| Optimizer | | AdamW | | |
| Hidden Layer Dimensions | 128:4 | 64:8 | 128 | 64 |
| LR Schedule | cosine | linear | cosine | cosine |
| LR Warmup | 0.1 | 0.05 | 0.1 | 0.1 |

Table 6: Hyperparameter used for reinforcement learning experiments.

| Hyperparameter | Llama3 1B | Aya 8B |
|---|---|---|
| Train Batch Size | | 128 |
| Optimizer | | AdamW |
| Max. Gradient Norm | | 1.0 |
| Learning Rate | | 0.000005 |
| LR Schedule | | constant |
| Weight Decay | | 0.0 |
| KL Weight | | 0.01 |
| Rollout Batch Size | | 128 |
| Rollouts per Step | | 128 |
| Rollout Temperature | | 1.0 |
| Rollout Min. Tokens | | 64 |
| Rollout Max. Tokens | | 1024 |
| Eval Temperature | | 1.0 |
| Eval Max. Tokens | | 1280 |
| Samples per Rollout | 32 | 8 |
| Max. Prompt Length | 1024 | 2560 |
| Frozen Layers | None | Image Encoder and lower half of LLM Layers |

## C    METADATA FOR MODEL TRAINING

We list the hyperparameter used for reward model training in Table 5. We ran a minimal grid search to find the best hidden dimensions, learning rate, schedule and warmup for the baseline ORM and MCQ-ORM. The hyperparameters were chosen based on the lowest loss on the development set. The hyperparameter for reinforcement learning experiments are listed in Table 6. They were chosen based on initial experiments with DrGRPO and Llama3 1B. The maximum prompt length for multimodal Aya is higher because it includes image tokens. Due to resource constraints, we set the number of samples to 8 and did not update the image encoder as well as the lower half (16) of the LLM layers.

## D    META EVALUATION

We use GPT-4.1 (OpenAI et al., 2024) with version *gpt-4.1-2025-04-14* as the judge to assess process correctness. To ensure future comparison and reproducibility of our results, we released the verbatim CoTs and the full assessment of the judge. The prompt we used is shown in Figure 9. We further conduct a meta evaluation that compares the LLM judge with human judgments and its ability to detect synthetically corrupted gold CoTs. Golovneva et al. (2023) released a dataset of 200 model generated CoTs for GSM8k together with human judgments of process correctness. We compare these human judgments with those from our LLM judge in Table 7. In 97% of the cases the human and LLM judge agree on the process correctness. Four of the six cases in which the human and judge disagree are shown in Figure 10, Figure 11, Figure 12 and Figure 13. It is up to the reader

Table 7: Human Meta Evaluation GSM8k. 97% overall.

| Human | #CoT | Judge Correct | Judge Incorrect | Judge Accuracy |
|-------|------|---------------|-----------------|----------------|
| Correct | 109 | 103 | 6 | 94.5% |
| Incorrect | 91 | 0 | 91 | 100% |

Table 8: Synthetic Meta Evaluation AQuA.

| Gold CoT | Corrupted CoT | #Count |
|----------|---------------|--------|
| Correct | Incorrect | 154 |
| Correct | Correct | 5 |
| Incorrect | Correct | 0 |
| Incorrect | Incorrect | 41 |

to decide whether the concerns by the judge are minor inaccuracies or invalidates the CoT. Next, we evaluate our LLM judge on a synthetic dataset. To this end, we use the gold rationals of AQuA and corrupt them by replacing a randomly selected number with a randomly generated number. Table 8 shows the judgment outcomes for the four possible cases. In the first row, we see the most desired outcome — the gold CoT is judged as correct and the corrupted CoT as incorrect. In only five cases, the LLM judge does not flag the corrupted CoT as incorrect. Because the LLM judge flags some gold rationals as incorrect, we manually inspect those CoTs. We find that all of them contain an error and show an example of an incorrect gold CoT in Figure 14.

---

Your task is to evaluate whether a reasoning process is fully correct and leads to the ground truth answer. I will present you a question, its ground truth answer and the reasoning process. Please analyze the reasoning and tell me if it is fully correct. Return your final assessment as \\boxed{{yes}} or \\boxed{{no}}. Remember, the presented answer to the question is the ground truth and you are solely judging the correctness of the reasoning process.

Question:
<>
Correct Answer:
<>
Reasoning:
<>

Is the reasoning process correct? Think step by step and return your final judgement as \\boxed{{yes}} or \\boxed{{no}}.

---

Figure 9: Prompt for LLM judge to assess the process correctness of a CoT.

## E  ORM ANSWER-ACCURACY

In addition to the Process-Accuracy (P-Acc) results in Table 1, we report the Answer-Accuracy (A-Acc) results in Table 9 for the reward model experiments. Here, the ORM baseline outperforms the MCQ-ORM method. However, in practice, one can use the ORM to select the CoT that is most likely answer-correct and subsequently the MCQ-ORM to select one of the answer-correct CoTs that is most likely process-correct.

## F  LLM USAGE

We used LLMs to rephrase single sentences, but not full paragraphs. We also used them to help with the correct formatting of equations and tables.

---

**Question:** The marching band is ordering new uniforms. Each uniform comes with a hat that costs $25, a jacket that costs three times as much as the hat, and pants that cost the average of the costs of the hat and jacket. How much does each uniform cost total?
**Gold Answer:** 150

**Chain-of-Thought:**
Step 1 - The jacket costs $25 * 3 = $<<25*3=75>>75.
Step 2 - The average cost of the hat and jacket is $25 + $75 = $<<25+75=100>>100.
Step 3 - The pants cost $100 / 2 = $<<100/2=50.0>>50.
Step 4 - So the total cost of each uniform is $25 + $75 + $50 = $<<25+75+50=150>>150
Step 5 - A: 150

**Human Label:** Correct

**Judge Label:** Incorrect
**Judge Reasoning:** Let's analyze the reasoning step by step:
Step 1: The jacket costs $25 * 3 = $75. - This is correct, as the jacket is three times the cost of the hat.
Step 2: The average cost of the hat and jacket is $25 + $75 = $100. - This is not the average; this is the sum. The average should be (25 + 75) / 2.
Step 3: The pants cost $100 / 2 = $50. - This step is correct in that it divides the sum by 2 to get the average, but the previous step mislabels the sum as the average.
Step 4: So the total cost of each uniform is $25 + $75 + $50 = $150. - This is correct, as it adds the costs of the hat, jacket, and pants.
Step 5: A: 150 - This is the correct answer.
However, in Step 2, the reasoning says "the average cost of the hat and jacket is $25 + $75 = $100," which is not the average but the sum. The average is calculated in the next step, but the labeling is incorrect.
Therefore, the reasoning process contains a misstatement about what the average is, even though the calculation ultimately arrives at the correct answer.
Final assessment: \\boxed{{no}}

Figure 10: Example of disagreement between human and LLM judgment.

Table 9: **Answer-Accuracy (A-Acc)** using different methods of CoT scoring. The task is to score multiple candidate CoTs. The highest scoring CoT is then evaluated for answer-correctness. This is done for all development set questions of the three multiple-choice QA datasets. Outcome-supervised reward models (ORM) are trained with three different random seeds and mean±std is reported. Nominal best values are bold.

|  | Llama3 1B | | | Llama3 8B | | |
|---|---|---|---|---|---|---|
|  | **AQuA** | **MATH** | **GSM8K** | **AQuA** | **MATH** | **GSM8K** |
| Random | 41.9 | 45.1 | 55.6 | 77.2 | 71.7 | 92.4 |
| Majority Voting | 53.1 | 56.4 | 73.2 | 88.5 | 81.4 | 98.1 |
| Answer Confidence | 59.9 | 59.8 | 82.7 | 88.9 | 83.3 | **98.3** |
| ORM | **66.3** ±0.1 | **63.8** ±0.3 | **86.8** ±0.4 | **89.4** ±0.1 | **83.7** ±0.1 | 98.2 ±0.1 |
| MCQ-ORM (ours) | 63.1 ±0.1 | 62.7 ±0.3 | 85.2 ±0.2 | **89.4** ±0.1 | 83.6 ±0.3 | **98.3** ±0.0 |

**Question:** Katy makes coffee using teaspoons of sugar and cups of water in the ratio of 7:13. If she used a total of 120 teaspoons of sugar and cups of water, calculate the number of teaspoonfuls of sugar she used.
**Gold Answer:** 42

**Chain-of-Thought:**
Step 1 - The total ratio representing the number of teaspoons of sugar and cups of water used is 7+13 = <<7+13=20>>20
Step 2 - The fraction representing the number of teaspoons of sugar used is 7/20, and since she used 120 teaspoons of sugar, she used 7/20*120 = <<7/20*120=42.0>>42 teaspoons of sugar
Step 3 - A: 42

**Human Label:** Correct

**Judge Label:** Incorrect
**Judge Reasoning:** Let's analyze the reasoning step by step:
Step 1: The total ratio representing the number of teaspoons of sugar and cups of water used is 7+13 = 20. This is correct. The total parts in the ratio are 7 (sugar) + 13 (water) = 20 parts.
Step 2: The fraction representing the number of teaspoons of sugar used is 7/20, and since she used 120 teaspoons of sugar, she used 7/20*120 = 42 teaspoons of sugar. Here is the issue: The statement says "since she used 120 teaspoons of sugar," but the problem says "she used a total of 120 teaspoons of sugar and cups of water." That is, 120 is the total of both ingredients, not just sugar. The reasoning should be: Number of teaspoons of sugar = (7/20) × 120 = 42.
The calculation is correct, but the reasoning misstates what the 120 represents. It should say "since she used a total of 120 teaspoons of sugar and cups of water," not "since she used 120 teaspoons of sugar."
Step 3: A: 42 The answer is correct.
However, because the reasoning misstates what the 120 represents, the reasoning process is not fully correct.
Final assessment: \\boxed{{no}}

Figure 11: Example of disagreement between human and LLM judgment.

1350
1351
1352
1353
1354
1355
1356
1357
1358
1359
1360
1361
1362
1363
1364
1365
1366
1367
1368
1369
1370
1371
1372
1373
1374
1375
1376
1377
1378
1379
1380
1381
1382
1383
1384
1385
1386
1387
1388
1389

---

**Question:** Lee used to be able to run the 400-meter hurdles two seconds faster than Gerald would run the 400-meter hurdles. But Gerald changed his diet, which improved his speed by 10%. If Lee runs the 400-meter hurdles in 38 seconds, how fast can Gerald, with his improved diet, run the 400-meter hurdles, in seconds?

**Gold Answer:** 36

**Chain-of-Thought:**
Step 1 - Since Lee is able to run the 400-meter hurdles two seconds faster than Gerald, then Gerald runs the 400-meter hurdles in 38 + 2 = <<38+2=40>>40 seconds.
Step 2 - If Gerald's speed improves by 10%, then his time to run the 400-meter hurdles is reduced by 10/100 * 40 = <<10/100*40=4.0>>4 seconds.
Step 3 - So, Gerald's time to run the 400-meter hurdles, after his diet improvements, is 40 - 4 = <<40-4=36>>36 seconds.
Step 4 - A: 36

**Human Label:** Correct

**Judge Label:** Incorrect
**Judge Reasoning:** Let's analyze the reasoning step by step:
Step 1: Lee runs the 400m hurdles in 38 seconds, which is 2 seconds faster than Gerald's old time. So Gerald's old time is 38 + 2 = 40 seconds. This is correct.
Step 2: Gerald's speed improves by 10%. The reasoning says that his time is reduced by 10% of 40 seconds, i.e., 4 seconds. This is a critical point.
- If speed increases by 10%, then the time to cover the same distance decreases, but not by 10%. The relationship between speed and time is inverse: time = distance / speed. - If original speed is S, and new speed is 1.1S, then new time is T' = D / (1.1S) = (1/1.1) * (D/S) = (1/1.1) * T ≈ 0.909 * T. - So, the new time should be 40 * (1/1.1) ≈ 36.36 seconds, not 36 seconds.
Step 3: The reasoning subtracts 4 seconds (10% of 40) from 40, getting 36 seconds. This is incorrect, as explained above.
Final assessment: The reasoning process is not fully correct. \\boxed{{no}}

---

Figure 12: Example of disagreement between human and LLM judgment.

1390
1391
1392
1393
1394
1395
1396
1397
1398
1399
1400
1401
1402
1403

**Question:** Candice put 80 post-it notes in her purse before she headed out to her job at the coffee shop. On her way, she stopped off at the store and purchased a package of Post-it notes; At work, she placed a single Post-it note on each of 220 different cups of coffee. If she had 23 post-it notes remaining overall, how many Post-it notes were in the package that she purchased?
**Gold Answer:** 163

**Chain-of-Thought:**
Step 1 - If she placed one post-it note on each of the 220 coffee cups, she used 220/1 = <<220/1=220.0>>220 post-it notes.
Step 2 - If she had 80 post-it notes in her purse before she purchased the package, then she used 220 - 80 = 140 post-it notes from the package.
Step 3 - If she had 23 post-it notes remaining, then the package contained 140 + 23 = <<140+23=163>>163 post-it notes.
Step 4 - A: 163

**Human Label:** Correct

**Judge Label:** Incorrect
**Judge Reasoning:** Let's analyze the reasoning step by step:
**Step 1:** "If she placed one post-it note on each of the 220 coffee cups, she used 220/1 = 220 post-it notes." - This is correct. She used 220 post-it notes.
**Step 2:** "If she had 80 post-it notes in her purse before she purchased the package, then she used 220 - 80 = 140 post-it notes from the package." - This is incorrect. She had 80 post-it notes, but after using 220, she would have a deficit of 140 post-it notes, not that she used 140 from the package. She could not have used 220 - 80 = 140 from the package, because she also had 23 left at the end.
**Step 3:** "If she had 23 post-it notes remaining, then the package contained 140 + 23 = 163 post-it notes." - This is also incorrect, because the calculation in Step 2 was incorrect. The correct way is to set up the equation: 80 + x (package) - 220 (used) = 23 (remaining), so 80 + x = 220 + 23, so x = 163.
**Step 4:** "A: 163" - The answer is correct, but the reasoning to get there is not.
**Conclusion:** The reasoning process is not fully correct, even though the final answer is correct.
Final assessment: \\boxed{{no}}

Figure 13: Example of disagreement between human and LLM judgment.

**Question:** A trader bought a car at 25% discount on its original price. He sold it at a 40% increase on the price he bought it. What percent of profit did he make on the original price?
**Options:** A. 118   B. 110   C. 105   D. 113   E. 115
**Gold Answer:** C
**Gold CoT:** Original price = 100 CP = 75 S = 75*(140/100) = 105 100 - 105= 5% Answer: C

**Judge Label:** Incorrect
**Judge Reasoning:** Let's analyze the reasoning step by step:
1. Original price = 100
This is a good assumption for calculation.
2. CP = 75
Since the trader bought at a 25% discount, the cost price (CP) is 75.
3. S = 75*(140/100) = 105
He sold it at a 40% increase on the price he bought it, so selling price (S) = 75 × 1.4 = 105.
4. 100 - 105 = 5%
This step is incorrect. The calculation "100 - 105 = 5%" is not the correct way to find the percent profit on the original price.
The correct way:
Profit = Selling Price - Original Price = 105 - 100 = 5 Profit percent on original price = (Profit / Original Price) × 100 = (5 / 100) × 100 = 5%
However, the answer is 5%, which matches option C (105%), but the reasoning is not clearly or correctly explained, especially in the last step.
Conclusion: The reasoning process contains a calculation error in the last step and does not clearly explain how the percent profit is calculated on the original price. Therefore, the reasoning is not fully correct.
\\boxed{{no}}

Figure 14: Gold rational of AQuA with error marked in red.

