# OpenReview forum: "Boosting Process-Correct CoT Reasoning by Modeling Solvability of Multiple-Choice QA"
_ICLR.cc/2026/Conference — Submitted to ICLR 2026_

### Official Review · Reviewer_FMkQ · 2025-10-30

**Soundness:** 3
**Presentation:** 1
**Contribution:** 2
**Rating:** 2
**Confidence:** 4

**Summary:**

The paper proposes to improve process-correctness in CoT reasoning by adapting outcome-based reward models and re-weighting group relative advantages during RL training. Using better than random performance as a heuristic for solvability, the authors derive a probability that a question is solvable for a particular model based on its observed group-wide performance. Under the assumption that outcome-correct generations for "unsolvable" problems are process-incorrect, this probability can be used to adapt outcome-based reward models and group-relative advantages for solution selection and RL respectively. The results indicate small improvements from the two approaches relative to the most similar baselines (standard ORM and Dr-GRPO) across two Llama3 models on the benchmarks considered.

**Strengths:**

The paper presents a novel and creative approach to incentivising process-correctness, which is an important problem. The results indicate that the approach has promise, despite being insufficient to be confident of its utility (see Weaknesses).

**Weaknesses:**

The paper needs considerable re-writing, as it is currently quite confusing to read. The limited introduction and unconventional following structure (method, background, more methods) makes it difficult to follow and prevents the paper from having a clear thread of motivation, approach and evidence.

The assumption that multiple choice problems with better than random performance are "solvable" and others are not seems restrictive and the connection between this assumption and true process-correctness is insufficiently evidenced. The results show modest gains over for example the typical Dr-GRPO advantage, but these alone, spanning only a limited number of benchmarks and two models from a single model family are insufficient to draw strong conclusions about the utility of p_solvable.

Figure 1 (right) shows the potential limitation of p_solvable, specifically that for low # Answer-Correct CoTs, p_solvable trails the fraction of at least one process-correct CoT, indicating that opportunities for learning from a process-correct CoT are discarded. This could come with the added risk of limiting exploration, which should also be investigated.

**Questions:**

How does re-weighting the advantage by p_solvable affect exploration?

What is the rate of process-correct CoT's for "unsolvable" problems and how do you propose to not waste these informative samples?

---

> ### Author Response · Authors · 2025-12-03
> **Response to Reviewer FMkQ**
>
> We thank the reviewer for their valuable feedback and suggestions.
>
> - Weakness 1 (paper structure):
> We take follow your advice about paper structure and rearrange the sections to background→general_method→method_application1→method_application2.
>
> - Weakness 2 (modest gains):
> We acknowledge that gains in answer-correctness and process-correctness are small, but want to emphasis that they are statistically significant (Appendix B) and consistent across experiments, training progress and random seeds. In Table 2, our method is nominal equal or better in 8/9 (3 dev sets, 3 random training seeds each) cases for A-Acc and 7/9 in P-Acc. In Table 3, these numbers are 11/12 and 9/12 respectively.
>
> - Weakness 3 (limited benchmarks):
> We ran additional experiments on the three math datasets using Qwen2.5 1.5B. The results are in line with previous results, further strengthening the usefulness of the proposed method. We want to highlight, that we test our method not only in the math domain that dominates previous reasoning literature, but also in a vision and language setting that is often under-explored. The used Aya VLM is not derived from Llama3, but uses “Command R” as the base LLM. After the added experiments we tested our method on 3 different base models, and 5 different datasets (3 math-related and 2 VQA).
>
> - Weakness 4/Question 1/Question 2 (limiting exploration):
> The reviewer is correct that the proposed method limits exploration to a certain degree, as discussed in the Analysis section and illustrated by Figure 5 (right). However, the accuracy evaluation shows that trading exploration for a higher probability of process-correct training examples leads to a model that has higher process- and answer-accuracy, making the trade-off worthwhile. Samples are never fully discarded but given a lower advantage as they would get under DrGRPO.

---

### Official Review · Reviewer_CfFQ · 2025-10-30

**Soundness:** 4
**Presentation:** 3
**Contribution:** 3
**Rating:** 6
**Confidence:** 3

**Summary:**

This paper proposes modeling question solvability to mitigate the issue of spurious chain-of-thought reasoning in large language models, where answers may be correct but the reasoning process is flawed. The authors incorporate solvability probabilities into the reward model (MCQ-ORM) and the rl advantage function (MCQ-DrGRPO), thereby down-weighting chance-correct samples from unsolvable questions.

**Strengths:**

This paper further explains the value of medium-difficulty questions through the concept of Learning Potential and leverages solvability analysis to intuitively reveal the origins of false positive CoTs, providing strong interpretability and insights.

**Weaknesses:**

1. The proposed notion of Learning Potential is demonstrated on math and relatively simple multimodal QA tasks. How well does this metric generalize to other domains, such as commonsense reasoning, dialogue, or symbolic tasks, where difficulty and solvability may behave differently?

2. The current approach is restricted to multiple-choice QA. If extended to open-ended generation tasks (e.g., free-form QA or long-form reasoning), how would the solvability framework be adapted or redefined?

**Questions:**

see weaknesses

---

> ### Author Response · Authors · 2025-12-03
> **Response to Reviewer CfFQ**
>
> We thank the reviewer for their valuable feedback and suggestions.
>
> - Weakness 1 (generalization to other domains):
> In Figure 3 we have results for the SFT experiment on medical QA and social QA that indicate the same pattern as on math reasoning datasets that are further investigated in the RL experiments.
>
> - Weakness 2 (open-ended questions):
> This is an important question and we added a paragraph in the conclusion to discuss this. While we do not propose a direct extension to open-ended QA, we refer to a possible direction by rephrasing open-ended QA as MCQA [Zhang et al., 2025].
>
>    - “Auditable-choice reframing unlocks RL-based verification for open-ended tasks”, Zhang et al., 2025, https://arxiv.org/abs/2511.02463

---

### Official Review · Reviewer_WMmh · 2025-11-01

**Soundness:** 2
**Presentation:** 3
**Contribution:** 3
**Rating:** 4
**Confidence:** 4

**Summary:**

This paper proposes to incorporate solvability signal into chain-of-thought reasoning. It first provides a controlled experiment on multiple-choice QA and indicates that CoT for unsolvable questions are more likely to be spurious. Then it incorporates that solvability information into both model inference and reinforcement training, showing consistent improvements.

**Strengths:**

Improve the reliability of Chain-of-thought reasoning is important, therefore I believe the studied problem would be helpful for the community;
Intuitively I like this idea and I believe if executed well, this idea could work and generalize to different tasks and model families.

**Weaknesses:**

1. My biggest concern is that it only tests on Llama model, which is a non-reasoning model with limited reasoning capacity. To strengthen the findings, evaluating on other LLMs with different sizes and reasoning models is necessary, especially Qwen-series.
2. Even on Llama, the observed improvements (Tables 2 and 3) are limited and in some cases not statistically significant. The authors should provide additional analyses such as qualitative examples, to illustrate where the proposed approach works well and where it fails.
3. In addition to RL, experiments based on SFT, such as rejection sampling, would also help provide a more comprehensive understanding of the proposed approach.

**Questions:**

whether and to what extent does your proposed reward signal robust to reward hacking?

---

> ### Author Response · Authors · 2025-12-03
> **Response to Reviewer WMmh**
>
> We thank the reviewer for their valuable feedback and suggestions.
>
> - Weakness 1 (model family):
> We want to highlight that we do not only evaluate on Llama3 but also on Aya, a VLM model based on the Command R LLM. We additionally added experiments on Qwen2.5 1.5B as suggested by the  reviewer. The results are in line with previous results, further strengthening the usefulness of the proposed method. Due to limited resources we cannot train larger models.
>
> - Weakness 2 (limited improvements):
> We acknowledge that gains in answer-correctness and process-correctness are small, but want to emphasis that they are statistically significant (Appendix B) and consistent across experiments, training progress and random seeds. In Table 2, our method is nominal equal or better in 8/9 (3 dev sets, 3 random training seeds each) cases for A-Acc and 7/9 in P-Acc. In Table 3, these numbers are 11/12 and 9/12 respectively.
>
> - Weakness 3 (rejection sampling):
> We prioritized RL experiments in favor of additional rejection sampling experiments and want to highlight that the SFT experiments in Figure 3 resemble a form of rejection sampling.

---

### Official Review · Reviewer_jVWz · 2025-11-03

**Soundness:** 3
**Presentation:** 3
**Contribution:** 3
**Rating:** 6
**Confidence:** 3

**Summary:**

This paper examines the correctness of processes in reasoning tasks. The core idea consists of three main steps. First, it defines the solvability of a multiple-choice question by estimating the probability p_solvable that a model can solve a problem, using a Beta posterior derived from multiple sampled accuracies, with a threshold above random guessing. Second, it incorporates p_solvable into an ORM as a soft label for positive examples, helping the model select the most process-correct CoT among multiple candidates at test time. Third, it integrates p_solvable into the advantage function of DrGRPO, forming MCQ-DrGRPO, which down-weights examples that are correct but likely have flawed reasoning processes. The authors further introduce a “learning potential” measure (LP =) novelty × solvability to explain which data points are most useful. Across mathematical and multimodal benchmarks, both MCQ-ORM and MCQ-DrGRPO achieve higher process correctness and also improve answer accuracy in reinforcement learning experiments.

**Strengths:**

* The authors formalize the empirical observation that “false-positive CoTs are more common on difficult problems” using the notion of solvability. The derivation is simple and reproducible.
* The method connects p_solvable to two common pipelines (e.g., test-time CoT selection and training-time RLVR) with minimal modifications and low computational overhead.
* The work covers three math MCQA datasets and two multimodal “geography/year guessing” datasets, reports both process and answer accuracy, and includes LLM-judge meta-evaluation and released outputs for reproducibility.
* MCQ-ORM consistently outperforms the standard ORM, while MCQ-DrGRPO outperforms DrGRPO across random seeds, with stronger gains in process accuracy (P-Acc).

**Weaknesses:**

* The paper assumes a random-guessing baseline 1/|choices| as the solvability threshold, which is reasonable for MCQs but not directly generalizable to open-ended generation tasks. A more detailed discussion on how the concept of solvability could be extended to general generative settings would strengthen the paper.
* In Section 4, the authors compare solvability-based ORM with the standard ORM and several confidence/faithfulness/CoPS metrics. However, in Section 5, the reinforcement learning experiments only compare DrGRPO variants. Including confidence- or uncertainty-based GRPO baselines would make the results more convincing.

**Questions:**

Currently, p_solvable is estimated from multiple sampled accuracies. Since open-source models can expose logits (and even many APIs provide top-n logits), could solvability be calculated directly from logit-based uncertainty measures (such as sequence entropy, logit margin, or temperature-calibrated confidence) to reduce sampling cost or improve estimation quality?

---

> ### Author Response · Authors · 2025-12-03
> **Response to Reviewer jVWz**
>
> We thank the reviewer for their valuable feedback and suggestions.
>
> - Weakness 1 (open-ended QA):
> This is an important question and we added a paragraph in the conclusion to discuss this. While we do not propose a direct extension to open-ended QA, we refer to a possible direction by rephrasing open-ended QA as MCQA [Zhang et al., 2025].
>    - “Auditable-choice reframing unlocks RL-based verification for open-ended tasks”, Zhang et al., 2025, https://arxiv.org/abs/2511.02463
>
> - Weakness 2 (other baselines for RL):
> We believe the direct comparison with DrGRPO clearly supports our hypothesis that the advantage values of “unsolvable” questions have to be down-weighted. Our method does not have any hyperparameters and works out-of-the-box for MCQA. We also support the downstream results with theoretical foundation and analysis experiments (Section 5.1.).
>
>
> - Question 1 (incorporating logit-based uncertainty):
> While there certainly is opportunity to incorporating logit-based uncertainty, one of the advantages of the proposed method is its simplicity and lack of hyperparameters.
> That is certainly an interesting extension and worth to be considered in future work. In this work we show that the general concept of solvability (estimated from answer accuracy) is key to reduce process-incorrect CoT reasoning.

---

### Meta-Review · Area_Chair_Dy1e · 2026-01-09

**Summary:**

This paper analyzed the effect of intermediate reasoning step to the producing correct answer in LLMs by estimating the solvability in MCQA questions. While the merits of the paper (e.g.,  research problem is important, intuitive idea) appreciated by reviewers, there are several key weaknesses that concern the reviewers (and myself). Although the authors partially addressed some of these concerns during the rebuttal, major weaknesses remain.

**Reviewer Concerns:**

After a quick examination of the paper and a careful review of all rebuttals, I found that I share the same concerns raised by the reviewers, as summarized below:
- One major concern echoed by the reviewers is the targeted task format of multiple-choice QA limited their conclusion scope. While the authors agree with the reviewers and discussed in the rebuttal and update paper, but direct experimental support is missing. I encourage I would like to address this issue further in their next version of paper to make their conclusion more robust.
- There are other concerns raised by reviewers, including limited baseline/base model comparison(e.g, only compare DrGRPO variants,  only tests on Llama model etc);  limited benchmarks (e.g., only demonstrated on math and relatively simple multimodal QA tasks). While the authors have partially addressed these points in the rebuttal (e.g., additional experiments on the three math datasets using Qwen2.5 1.5B), I encourage them to incorporate these analyses into the paper in subsequent revisions.

Addressing the issues outlined above would make the paper impactful and I encourage the authors to integrate these additional content into the next version of the paper as it will strengthen their contribution more.

**Reviewer Scores:**

I do not think the reviewers would have changed their scores as their major concerns mentioned above (also mine) still remain.

---

### Decision · Program_Chairs · 2026-01-26

Reject